# Phenotypic and genotypic differences between bloodstream and non-blood *Candida krusei* isolates: implications for invasiveness and antifungal susceptibility

Ying Zhao,[1,2] Han Wang,[1,2] Jinhan Yu,[1] Yi Li,[1] Ge Zhang,[1] Wei Kang,[1] Meng Xiao,[1] Qiwen Yang,[1] Lina Guo,[1] Yingchun Xu[1]

**ABSTRACT**   *Candida krusei* is an opportunistic pathogen with intrinsic resistance to fluconazole. This study analyzed 174 *C. krusei* isolates from bloodstream infections (BSI, $n = 115$) and non-blood sources ($n = 59$). Biochemical profiling using the VITEK 2 YST ID Card revealed a significant difference in the TyrA reaction between BSI and non-blood isolates ($P < 0.05$). Antifungal susceptibility testing via CLSI broth microdilution and Sensititre YeastOne showed that BSI isolates were more susceptible to echinocandins and azoles than non-blood strains, with minor MIC discrepancies between methods. Microsatellite genotyping at eight loci demonstrated high genetic diversity. Notably, a cluster of respiratory isolates from COVID-19 patients suggested potential nosocomial transmission. The integration of phenotypic, susceptibility, and genotypic data highlights possible virulence traits and epidemiological patterns. These findings may support the development of tailored antifungal therapies and improved infection control, particularly for high-risk populations such as ICU and COVID-19 patients.

**IMPORTANCE** Bloodstream infections caused by *C. krusei* are associated with high mortality rates and limited therapeutic options, emphasizing the need to better understand the factors contributing to its pathogenicity and invasive potential. This study provides critical insights into the biology and clinical impact of *C. krusei*. By integrating biochemical, antifungal susceptibility, and genotypic analyses of blood-stream and non-bloodstream isolates, it reveals distinct phenotypic and resistance profiles, high genetic diversity, and potential nosocomial transmission in COVID-19 patients. These findings underscore the importance of recognizing population-specific traits in *C. krusei*, which may guide tailored antifungal strategies and strengthen infection control measures, especially for vulnerable groups such as ICU and immunocompromised patients.

**KEYWORDS**   *Candida krusei*, biochemical profiling, antifungal susceptibility, microsatellite typing

C*andida krusei* (sexual form *Pichia kudriavzevii*) is an opportunistic fungal pathogen that poses significant clinical challenges due to its intrinsic resistance to flucona-zole and its increasing prevalence in nosocomial infections. Although less frequently isolated than *C. albicans* or *C. glabrata*, *C. krusei* has been recognized as a notable cause of bloodstream infections (candidemia), particularly among immunocompromised and critically ill patients. Bloodstream infections caused by *C. krusei* are associated with high mortality rates and limited therapeutic options, emphasizing the need to better understand the factors contributing to its pathogenicity and invasive potential. Its capacity for persistent colonization of mucosal surfaces and skin, as well as potential for nosocomial spread, raises significant concerns in hospital environments. The possibility

**Peer Reviewer** Ehab A. Salama, Virginia Polytechnic Institute and State University, Blacksburg, Virginia, USA

Address correspondence to Ying Zhao, zhaoying28062806@163.com, Lina Guo, guo0201205@126.com, or Yingchun Xu, xycpumch@139.com.

The authors declare no conflict of interest.

See the funding table on p. 14.

of clonal transmission of *C. krusei* within healthcare settings has increasingly been recognized. *C. krusei* candidemia outbreak is reported very infrequently both in adults and pediatric population across the globe (1–4).

*C. krusei* can also be isolated from non-invasive sites, including the respiratory tract, urine, and gastrointestinal tract, where it may act as a colonizer or cause localized infections (5, 6). Whether bloodstream isolates possess specific characteristics that distinguish them from clinical non-blood isolates remains an important but insufficiently explored question. Investigating such differences can help identify factors associated with the transition from colonization to invasive infection.

Biochemical profiling enables the comparison of metabolic capacities that may reflect adaptations to different host environments. *In vitro* antifungal susceptibility testing provides essential information on resistance patterns that may influence treatment outcomes and reflect strain-level differences in antifungal tolerance. Moreover, genotyping techniques such as microsatellite typing offer high-resolution insights into the genetic diversity and relatedness of clinical isolates, enabling the detection of possible clonal dissemination or genotype-specific pathogenic traits.

Recent studies have demonstrated the utility of microsatellite typing in assessing the genetic diversity of *C. krusei* isolates. For instance, Karakoyun et al. (4) analyzed *C. krusei* isolates from bloodstream and vaginal samples using short tandem repeat (STR) analysis, revealing unique genotypes and no evidence of clonal transmission. Similarly, Gong et al. (7) identified significant genetic differentiation among *C. krusei* isolates and observed variations in antifungal susceptibility profiles.

By comprehensively analyzing the phenotypic traits, antifungal susceptibility profiles, and microsatellite genotypes of *C. krusei* isolates from bloodstream and non-blood clinical specimens, this study aims to uncover potential invasiveness-associated factors. Elucidating such differences is crucial for understanding the pathobiology of *C. krusei*, improving early recognition of high-risk strains, and informing targeted therapeutic and infection control strategies in clinical settings.

## MATERIALS AND METHODS

### Isolate collection and identification

A retrospective study was conducted on a total of 174 strains. Among these strains, 122 non-duplicate *C. krusei* isolates cultured from 116 patients with Invasive Candidiasis (IC) were collected from 53 hospitals in the National China Hospital Invasive Fungal Surveillance Net (CHIF-NET) Study program 2009 and 2021. Basic patients and sampling information were collected, while detailed clinical data were not included in this retrospective investigation. Multiple clinical isolates from the same patient but different specimen types or different sample collection times were included. Among the 116 patients, there were 2 patients who had isolates cultured simultaneously from peripheral blood and central venous catheter blood, 3 patients who had 2 isolates cultured from peripheral blood collected at different dates (6 days, 18 days, and 30 days apart, respectively), 1 patient who had 3 isolates cultured from peripheral blood collected at different dates (the specimen collecting time between the first and second isolates was 8 days, and the second and third isolates was 12 days) (Table 1). And the remaining 52 non-duplicate *C. krusei* isolates were collected from 52 patients at Peking Union Medical College Hospital (PUMCH) during 2022–2024, a large tertiary hospital that treats diverse patient populations at risk for candidemia, including ICU patients, neonates, and immunocompromised individuals.

All strains were forwarded to the central laboratory (The Department of Clinical Laboratory, Peking Union Medical College Hospital) for confirmation of species identification by using the Vitek MS system (bioMerieux, Marcy l'Etoile, France) selectively supplemented by ribosomal DNA sequencing as required (8). The program was approved by the Human Research Ethics Committee of Peking Union Medical College Hospital (S-263 and 1-23PJ1093).

**TABLE 1** Multiple clinical isolates from the same patient but different specimen types or sample collection times were included[a,b]

| Patient No. | Province | Hospital | Age | Gender | Diagnosis | Isolate No. | Department_Category | Specimen collection date | Source | MT type | FZ | VOR | IZ | PZ | CAS | MF | AND | FC | AB |
|---|---|---|---|---|---|---|---|---|---|---|---|---|---|---|---|---|---|---|---|
| Patient 1 | Heilongjiang | H4 | 79 | Female | Chronic renal failure | K009 | Inpatient_ICU | 2/9/2012 | Central venous catheter blood | MT087 | 16 | 0.125 S | 0.5 WT | 0.5 WT | 0.5 I | 0.25 S | 0.06 S | 32 | 2 WT |
| | | | | | | K010 | Inpatient_ICU | 2/9/2012 | Peripheral blood | MT086 | 16 | 0.125 S | 0.25 WT | 0.5 WT | 0.5 I | 0.25 S | 0.12 S | 16 | 2 WT |
| Patient 2 | Beijing | PU | 66 | Female | Acute leukemia | K014 | Inpatient_Medical | 12/10/2012 | Peripheral blood | MT085 | 8 | 0.06 S | 0.06 WT | 0.25 WT | 0.5 I | 0.25 S | 0.12 S | 8 | 1 WT |
| | | | | | | K015 | Inpatient_Medical | 12/16/2012 | Peripheral blood | MT085 | 32 | 0.125 S | 1 WT | 0.5 WT | 0.25 S | 0.25 S | 0.12 S | 32 | 2 WT |
| Patient 3 | Beijing | GH | 32 | Female | Ruptured blood vessels in the neck, bleeding; Right hemothorax; Subcutaneous hematoma in the neck; After left thyroid nodule puncture | K018 | Inpatient_ICU | 8/29/2012 | Catheter blood | MT011 | 16 | 0.125 S | 0.5 WT | 0.5 WT | 0.5 I | 0.25 S | 0.12 S | 32 | 2 WT |
| | | | | | | K028 | Inpatient_ICU | 8/29/2012 | Peripheral blood | MT011 | 16 | 0.125 S | 0.5 WT | 0.25 WT | 0.25 S | 0.25 S | 0.12 S | 32 | 1 WT |
| Patient 4 | Heilongjiang | H1 | 24 | Male | Trauma | K039 | Inpatient_ICU | 2/25/2015 | Peripheral blood | MT073 | 16 | 0.125 S | 0.5 WT | 0.25 WT | 0.25 S | 0.125 S | 0.12 S | 32 | 2 WT |
| | | | | | | K040 | Inpatient_Medical | 3/5/2015 | Peripheral blood | MT077 | 16 | 0.125 S | 0.25 WT | 0.25 WT | 0.25 S | 0.5 I | 0.12 S | 32 | 2 WT |
| | | | | | | K041 | Inpatient_Medical | 3/17/2015 | Peripheral blood | MT077 | 32 | 0.25 S | 0.5 WT | 0.25 WT | 0.25 S | 0.25 S | 0.12 S | 16 | 2 WT |
| Patient 5 | Henan | HN | 49 | Female | FUO; After coronary heart disease bypass surgery; Diabetes and hypertension | K093 | Emergency | 1/10/2019 | Peripheral blood | MT094 | 16 | 0.125 S | 0.5 WT | 0.5 WT | 0.5 I | 0.25 S | 0.25 S | 32 | 2 WT |
| | | | | | | K094 | Inpatient_Medical | 1/28/2019 | Peripheral blood | MT094 | 16 | 0.25 S | 0.5 WT | 0.5 WT | 0.5 I | 0.25 S | 0.06 S | 32 | 2 WT |
| Patient 6 | Henan | HN | 71 | Male | FUO: lung infection? Acute cerebral infarction; Postoperative cholangiocarcinoma; Hypertension; Type 2 diabetes | K096 | Inpatient_ICU | 9/15/2019 | Peripheral blood | MT023 | 16 | 0.25 S | 0.25 WT | 0.5 WT | 0.5 I | 0.25 S | 0.12 S | 32 | 2 WT |
| | | | | | | K097 | Inpatient_ICU | 10/15/2019 | Peripheral blood | MT120 | 16 | 0.125 S | 0.5 WT | 0.5 WT | 0.25 S | 0.25 S | 0.5 I | 32 | 2 WT |

[a]Antifungal susceptibility testing results by CLSI method.
[b]FZ, fluconazole; VOR, voriconazole; IT, itraconazole; PZ, posaconazole; CAS, caspofungin; MF, micafungin; AND, anidulafungin; FC, flucytosine; AB, amphotericin B; S, susceptible; R, resistant; I, intermediate; WT, wild type; NWT, non-wild type. MIC values of AND, MF, CAS, VOR, and FZ were interpreted according to breakpoints in CLSI-M27M44S and ECVs in CLSI-M57S. FUO, fever of unknown origin.

## Biochemical profiling

Biochemical profile of the strains was obtained with automated Vitek 2 system (bioMérieux, Durham, USA) using VITEK 2 YST Identification Card (bioMérieux, Marcy l'Etoile, France) according to the manufacturer's instructions. In short, pure subcultures of the strains were suspended in aqueous 0.45% (wt/vol) NaCl to achieve a turbidity equivalent to that of a McFarland 2.0 standard (range, 1.80 to 2.20), as measured by the DensiCHE-Kautomated McFarland measurement (bioMérieux, Marcy l'Etoile, France). The VITEK 2 instrument automatically filled, sealed, and incubated the individual test cards with the prepared culture suspension. Cards were held at 35.5°C for about 18 h, with optical readings taken automatically every 15 min. Based on these readings, an identification profile was established and interpreted according to a specific algorithm.

## Antifungal susceptibility testing

*In vitro* antifungal susceptibility to nine antifungal drugs was performed both using the Sensititre YeastOne YO10 system (Thermo Scientific, Cleveland, OH, United States) following the manufacturer's instructions (SYO) and according to the CLSI document M27-A3 broth microdilution method (CLSI). The quality control strains included *Candida parapsilosis* ATCC 22019 and *C. krusei* ATCC 6258. MIC values of anidulafungin, micafungin, caspofungin, voriconazole, and fluconazole were interpreted according to breakpoints in CLSI-M27M44S (9) and the epidemiological cut-off values (ECVs) in CLSI-M57S (10). Where there are no CLSI breakpoints, species-specific epidemiology cutoff values (ECVs) were used to define isolates as wild type (WT) or non-wild type (NWT) for amphotericin B, posaconazole, itraconazole. To determine agreement between SYO and CLSI, use minimal inhibitory concentration (MIC) and interpreted results for calculation. Essential agreement (EA) was defined in terms of discrepancies in MIC results of no more than ±2-fold dilutions between SYO and CLSI; category agreement (CA) was defined agreement of susceptible, intermediate, and resistant results between SYO and CLSI method; major error (ME) was classified as results of resistance to SYO and susceptibility to CLSI; very major errors (VME) were classified as results of susceptibility to SYO and resistance to CLSI; minor errors (mE) occurred when the result of one of the tests was susceptible or resistant and that of the other test was intermediate (11). For all nine antifungal agents, we assessed EA rates between the SYO and CLSI methods. For antifungal agents with established clinical breakpoints (i.e., Voriconazole, Caspofungin, Anidulafungin, and Micafungin), we assessed CA, ME, mE, and VME rates between the SYO and CLSI methods.

## DNA extraction and microsatellite typing

After culturing isolates on SDA for 2 days, the genomic DNA was extracted using the Fungi Genomic DNA Extraction Kit (Solarbio Science & Technology, Beijing, China) according to the company's recommended protocols. Genotyping of all *C. krusei* isolates was performed using a panel of 8 highly polymorphic microsatellite markers as described by Jie Gong et al., namely, Cakr004, Cakr005, Cakr011, Cakr019, Cakr025, Cakr026, Cakr029, Cakr031. Amplification reactions and allele sizes analysis were performed as previously reported. Briefly, the PCR amplification process was started with initial denaturation at 94°C for 5 min, followed by 35 cycles of denaturation at 94°C for 30 s, annealing at 60°C for 45 s, extension at 72°C for 50 s, and final extension at 72°C for 5 min. PCR products were first subjected to agarose gel electrophoresis for quality control. Qualified samples were further analyzed using capillary electrophoresis. A mixture of formamide and molecular weight marker (100:1 vol ratio) was prepared. Fifteen microliters of this mixture was added to the loading plate, followed by 1 µL of PCR product diluted 10-fold. Capillary electrophoresis was performed using the 3730XL ABI sequencer (Applied Biosystems, Carlsbad, USA). The raw data obtained from the sequencer were analyzed using Genemarker V2.2.0 software with the Fragment (Plant) module. The molecular weight marker positions in each lane were compared to the peak positions of the samples to determine fragment sizes. The genetic relationships

between the genotypes were studied by constructing a minimum spanning tree using the BioNumerics software v7.6 (Applied Maths, Sint-Martens-Latem, Belgium), treating the data as categorical information. Genotypes showing the same alleles for all 8 markers were considered identical. Endemic genotypes were defined as genotypes infecting two different patients. A cluster was defined as a group of two patients infected by an endemic genotype (12).

Phylogenetic relationships were inferred using MEGA (version 10.2.6) software. A neighbor-joining (NJ) tree was constructed based on the proportion of shared alleles or p-distance, with 1,000 bootstrap replicates applied to assess branch reliability. The resulting phylogenetic tree was exported in Newick (.nwk) format. The Newick file was uploaded to the Interactive Tree of Life (iTOL) platform (https://itol.embl.de) to facilitate high-resolution visualization and annotation.

## RESULTS

### Clinical characteristics of patients

One hundred and fifteen isolates were isolated from blood or catheter tip specimens of 111 patients, ages ranging from 6 months to 91 years old, 65 males, 46 females. 19 cases of hematologic malignances, including lymphoma, leukemia, multiple myeloma. Twenty-five cases of esophageal cancer, liver tumors, abdominal trauma, peritonitis, etc. Eighteen cases of septicemia, fever of unknown origin, head and face skin infections, endocarditis, and other infectious diseases.

Other 59 isolates were cultured from vaginal swab specimens ($n$ = 19), tissue specimens ($n$ = 10), respiratory specimens ($n$ = 9) (include tracheal bronchial aspirate [$n$ = 3], bronchoalveolar lavage (BAL) [$n$ = 3], sputum [$n$ = 2], and tracheostomy secretion [$n$ = 1]), drainage fluid specimens ($n$ = 7), urine ($n$ = 6), ascites ($n$ = 2), wound secretion ($n$ = 2), stool ($n$ = 2), pus ($n$ = 1), oral swab ($n$ = 1). All isolates information were in the supplementary materials (Table S1）

### Biochemical profiling

To better visualize the differences in biochemical reactions between the two groups, a heatmap was plotted using R (v4.2.1; https://cran.r-project.org/) (Fig. 1). Among the 46 biochemical reactions tested with VITEK 2 YST ID Card, only Tyrosine arylamidase (TyrA) activity showed a statistically significant difference in positivity rates between *C. krusei* isolates from bloodstream (including catheter tip) samples and those from non-blood specimens. Specifically, 46.1% (53/115) of bloodstream isolates were positive for TyrA, compared to 76.3% (45/59) of non-blood isolates ($P$ = 0.00027). This suggests a potential phenotypic divergence related to invasiveness. Other biochemical reactions—including urease (URE), xylitol assimilation (XLTa), glycerol assimilation (GLYLa), and others—did

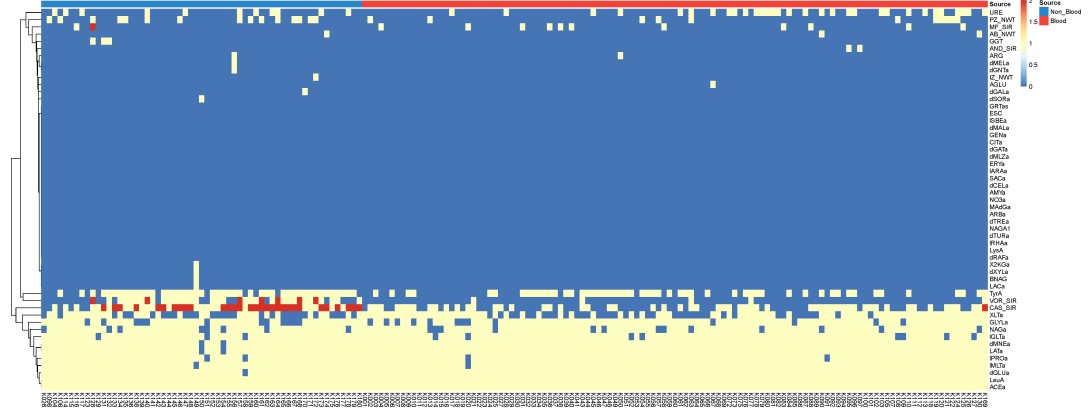

**FIG 1** Heatmap was plotted using R (v4.2.1; https://cran.r-project.org/) to visualize the differences in biochemical reactions and antifungal susceptibility between isolates from bloodstream and non-blood clinical specimens.

not show statistically significant differences between the two groups (all *P* > 0.05) although xylitol assimilation showed a borderline *P*-value (*P* = 0.074).

## Antifungal susceptibility of *C. krusei* isolates

The antifungal susceptibilities of the isolates are shown in Table 2. In general, the overall sensitivity of *C. krusei* in this study to itraconazole and posaconazole exhibit strong, with low resistance rates. For voriconazole shows moderate resistance, with higher resistance rates in non-blood specimens. For echinocandins (caspofungin, anidulafungin, and micafungin) have low MIC values, indicating high efficacy, though caspofungin shows some resistance. Amphotericin B has relatively consistent MIC values, suggesting good susceptibility. 5-Fluorocytosine displays variable MIC values, with some resistance noted.

There were some differences in MIC values when comparing these two antifungal susceptibility testing methods: CLSI and SYO. Overall, EA% (all >95%) was very high across all antifungals tested. The CA values ranged widely, with anidulafungin showing the highest agreement (97.70%) and caspofungin the lowest (55.17%). The relatively low categorical agreement (CA) observed for certain antifungal agents—particularly caspofungin and posaconazole—may be attributed to the fact that a substantial proportion of isolates had MIC values near the established clinical breakpoints. Importantly, all ME rates and VME rates were below 2% for the analyzed four antifungal agents. SYO generally reported slightly higher MIC50 and MIC90 values for some drugs, such as voriconazole, where SYO recorded an MIC50 of 0.5 µg/mL, while CLSI was 0.25 µg/mL. In contrast, for amphotericin B, SYO tend to give slightly lower MIC values than CLSI. The resistance rates for some drugs were slightly higher using SYO. For instance, posaconazole had a resistance rate of 20.6% in SYO vs 14.9% in CLSI. However, for most drugs, both methods yield comparable resistance percentages. The susceptibility rates were nearly identical for drugs like itraconazole and anidulafungin. However, for voriconazole and caspofungin, SYO shows slightly higher susceptibility rates compared to CLSI.

We also found there were some differences in MIC values and resistance rates among different specimen types. Blood and catheter specimens generally had lower MIC50 and MIC90 values. For example, voriconazole had an MIC50 of 0.125 µg/mL in blood samples vs 1 µg/mL in other specimens. Isolates from blood samples were more susceptible to antifungal drugs, with itraconazole, voriconazole, and posaconazole showing susceptibility rates above 90% in blood specimens, compared to less than 75% in other specimens. Voriconazole showed a resistance rate of 11.7% in other specimens, while in blood samples, it was 0%. Caspofungin and posaconazole also had higher MIC values in non-blood samples. A comparison of antifungal non-susceptibility (NS) /non-wildtype (NWT) rates between *C. krusei* isolates from bloodstream (including catheter tip) and non-blood specimens revealed significant differences for specific azole agents. Notably, voriconazole (VOR) resistance was substantially more common among non-blood isolates, with 62.7% (37/59) exhibiting non-susceptibility compared to only 4.3% (5/115) in bloodstream isolates (*P* < 0.0001). Similarly, posaconazole (POS) non-susceptibility was higher in non-blood isolates (25.4% vs 9.6%, *P* = 0.011). No statistically significant differences were found in non-susceptibility rates for caspofungin (CAS), micafungin (MF), anidulafungin (AND), amphotericin B (AMB), or itraconazole (IT) between the two groups (*P* > 0.05 for all).

## Genetic diversity based on microsatellite typing of *C. krusei* isolates

We found high genetic diversity among the 174 *C. krusei* isolates studied, a total of 145 different MT types containing 1–7 isolates were identified (Table S2). Most isolates had two or three alleles per marker, indicating that these were heterozygous diploid or triploid strains. Among the 145 MT types, 131 genotypes were unique and isolated from one patient each ; the remaining 14 were endemic and formed 14 clusters identified (named 1 to 14) that involved 40 patients (2 to 7 patients per cluster) (Table 3, Fig. 2).

**TABLE 2** Antifungal susceptibility testing results of all the clinical C. krusei isolates in this study[a]

| Antifungal name | MIC interpretive categories (µg/mL) | | All specimen types (n = 174) | | | | Agreement/error | | | | | Blood and catheter (n = 115) | | | | Other specimen types (n = 59) | | | |
|---|---|---|---|---|---|---|---|---|---|---|---|---|---|---|---|---|---|---|---|
| | | | R/NWT% | S/WT% | MIC50 | MIC90 | EA% | CA% | mE% | ME% | VME% | R/NWT% | S/WT% | MIC50 | MIC90 | R/NWT% | S/WT% | MIC50 | MIC90 |
| Fluconazole | IR | CLSI | –[b] | – | 32 | 128 | – | – | – | – | – | – | – | 16 | 32 | – | – | 128 | 256 |
| | | SYO | – | – | 64 | 64 | 95.40 | – | – | – | – | – | – | 64 | 64 | – | – | 32 | 64 |
| Itraconazole | ECV 1 | CLSI | 0.6 | 99.4 | 0.5 | 1 | – | – | – | – | – | 0 | 100 | 0.5 | 1 | 1.7 | 98.3 | 0.5 | 1 |
| | | SYO | 0.6 | 99.4 | 0.5 | 0.5 | 98.85 | – | – | – | – | 0 | 100 | 0.25 | 0.5 | 1.7 | 98.3 | 0.5 | 0.5 |
| Voriconazole | S <= 0.5 I 1 R >= 2 | CLSI | 4 | 75.4 | 0.25 | 1 | – | – | – | – | – | 0 | 95.7 | 0.125 | 0.25 | 11.7 | 36.7 | 1 | 2 |
| | | SYO | 2.3 | 89.1 | 0.5 | 1 | 95.40 | 75.86 | 21.26 | 1.15 | 1.72 | 1.7 | 93 | 0.25 | 0.5 | 3.3 | 81.7 | 0.5 | 1 |
| Posaconazole | ECV 0.5 | CLSI | 14.9 | 85.1 | 0.5 | 1 | – | – | – | – | – | 9.6 | 90.4 | 0.5 | 0.5 | 25 | 75 | 0.5 | 1 |
| | | SYO | 20.6 | 79.4 | 0.5 | 1 | 98.85 | – | – | – | – | 4.3 | 95.7 | 0.25 | 0.5 | 51.7 | 48.3 | 1 | 2 |
| Caspofungin | S <= 0.25 R >= 1 | CLSI | 18.3 | 36 | 0.5 | 1 | – | – | – | – | – | 0.9 | 49.6 | 0.5 | 0.5 | 51.7 | 10 | 1 | 1 |
| | | SYO | 22.9 | 53.1 | 0.25 | 1 | 100.00 | 55.17 | 44.83 | 0.00 | 0.00 | 2.6 | 70.4 | 0.25 | 0.5 | 61.7 | 20 | 1 | 1 |
| Anidulafungin | S <= 0.25 R >= 1 | CLSI | 0 | 98.9 | 0.125 | 0.125 | – | – | – | – | – | 0 | 98.3 | 0.064 | 0.125 | 0 | 100 | 0.125 | 0.125 |
| | | SYO | 0 | 92 | 0.125 | 0.25 | 97.13 | 97.70 | 2.30 | 0.00 | 0.00 | 0 | 99.1 | 0.125 | 0.125 | 0 | 78.3 | 0.25 | 0.5 |
| Micafungin | S <= 0.25 R >= 1 | CLSI | 0.6 | 92.6 | 0.25 | 0.25 | – | – | – | – | – | 0 | 90.4 | 0.25 | 0.25 | 1.7 | 96.7 | 0.25 | 0.25 |
| | | SYO | 0 | 98.8 | 0.064 | 0.25 | 100.00 | 84.48 | 14.94 | 0.00 | 0.57 | 0 | 100 | 0.064 | 0.125 | 0 | 96.7 | 0.125 | 0.25 |
| Amphotericin B | ECV 2 | CLSI | 1.7 | 98.3 | 2 | 2 | – | – | – | – | – | 1.7 | 98.3 | 2 | 2 | 1.7 | 98.3 | 2 | 2 |
| | | SYO | 4 | 96 | 1 | 2 | 99.43 | – | – | – | – | 0.9 | 99.1 | 1 | 2 | 10 | 90 | 2 | 2 |
| 5-Fluorocytosine | N/A | CLSI | – | – | 16 | 32 | – | – | – | – | – | – | – | 32 | 32 | – | – | 16 | 32 |
| | | SYO | – | – | 16 | 32 | 97.13 | – | – | – | – | – | – | 8 | 16 | – | – | 32 | 32 |

[a]S, susceptible; R, resistant; I, intermediate; WT, wild type; NWT, non-wild type. N/A, not applicable. MIC interpretive categories according to breakpoints in CLSI-M27M44S and ECVs in CLSI-M57S.
[b]"–" indicates not applicable.

The patients involved in 3 of the 14 clusters (21%) were geographically related. There was one clustering of genotypes involved patients admitted to the same ward at the time of sample collection, in the ICU wards of a hospital in Beijing in 2023, these five isolates from five different patients' respiratory specimens shared identical genotype (Table 4). On iTOL, metadata of sample type (bloodstream vs non-blood) were mapped onto the tree using colored rings (Fig. 3), which demonstrated that isolates from bloodstream vs non-blood specimens were genetically associated and distributed across a wide range of genetic clusters.

We analyzed the genotype of the multiple clinical isolates from the same patient but different specimen types or sample collection times. In one patient, genotypes were different between the isolates from peripheral blood and central venous catheter blood samples, the genotypes differed in only one marker. Whereas in another patient, isolates

TABLE 3  Fourteen clusters found in microsatellite typing

| Cluster | MT type | No. of patients involved | Isolate No. | Specimen type | Gender | Province | Hospital | Year |
|---|---|---|---|---|---|---|---|---|
| 1 | MT001 | 2 | K078 | Blood | F | Sichuan | HX | 2018 |
| | | | K081 | Blood | M | Jiangsu | NJ | 2018 |
| 2 | MT005 | 2 | K012 | Catheter | F | Jiangsu | JS | 2012 |
| | | | K013 | Blood | F | Beijing | PU | 2012 |
| 3 | MT007 | 2 | K101 | Blood | M | Shanghai | CH | 2020 |
| | | | K118 | Blood | M | Guizhou | GY | 2021 |
| 4 | MT008 | 2 | K030 | Blood | M | Liaoning | LR | 2014 |
| | | | K088 | Blood | F | Hunan | Y2 | 2018 |
| 5 | MT016 | 2 | K086 | Blood | M | Hunan | XY | 2018 |
| | | | K104 | Tissue | M | Guizhou | GY | 2020 |
| 6 | MT017 | 7 | K044 | Blood | F | Jiangsu | JS | 2015 |
| | | | K055 | Blood | M | Shandong | QH | 2016 |
| | | | K064 | Blood | M | Beijing | PU | 2017 |
| | | | K069 | Blood | F | Hunan | Y2 | 2017 |
| | | | K083 | Catheter blood | M | Shandong | QH | 2018 |
| | | | K085 | Blood | M | Beijing | TR | 2018 |
| | | | K113 | Blood | M | Chongqing | CQ | 2021 |
| 7 | MT018 | 2 | K035 | Blood | F | Henan | ZZ | 2014 |
| | | | K046 | Catheter blood | M | Jiangsu | SZ | 2015 |
| 8 | MT054 | 3 | K033 | Blood | M | Liaoning | Z1 | 2014 |
| | | | K114 | Tissue | M | Guangdong | G1 | 2021 |
| | | | K115 | Tissue | F | Guangdong | G1 | 2021 |
| 9 | MT061 | 2 | K026 | Lung tissue | M | Hunan | XY | 2013 |
| | | | K119 | Blood | M | Neimenggu | NM | 2021 |
| 10 | MT068 | 2 | K021 | Blood | M | Jiangsu | JS | 2013 |
| | | | K092 | Blood | F | Guangxi | GX | 2019 |
| 11 | MT083 | 2 | K125 | Blood | M | Hunan | XY | 2021 |
| | | | K126 | Blood | M | Hunan | Y2 | 2021 |
| 12 | MT084 | 3 | K024 | Blood | F | Zhejiang | WZ | 2013 |
| | | | K051 | Blood | M | Guangdong | GZ | 2016 |
| | | | K059 | Blood | M | Zhejiang | ZD | 2016 |
| 13 | MT085 | 3 | K014 | Blood | F | Beijing | PU | 2012 |
| | | | K015 | Blood | F | Beijing | PU | 2012 |
| | | | K016 | Blood | F | Shanghai | RJ | 2012 |
| | | | K108 | Blood | M | Gansu | LZ | 2020 |
| 14 | MT110 | 5 | K143 | Tracheal bronchial aspirate | M | Beijing | PU | 2023 |
| | | | K145 | Tracheal bronchial aspirate | M | Beijing | PU | 2023 |
| | | | K147 | BALF | M | Beijing | PU | 2023 |
| | | | K154 | Tracheal bronchial aspirate | M | Beijing | PU | 2023 |
| | | | K157 | Tracheostomy secretions | M | Beijing | PU | 2023 |

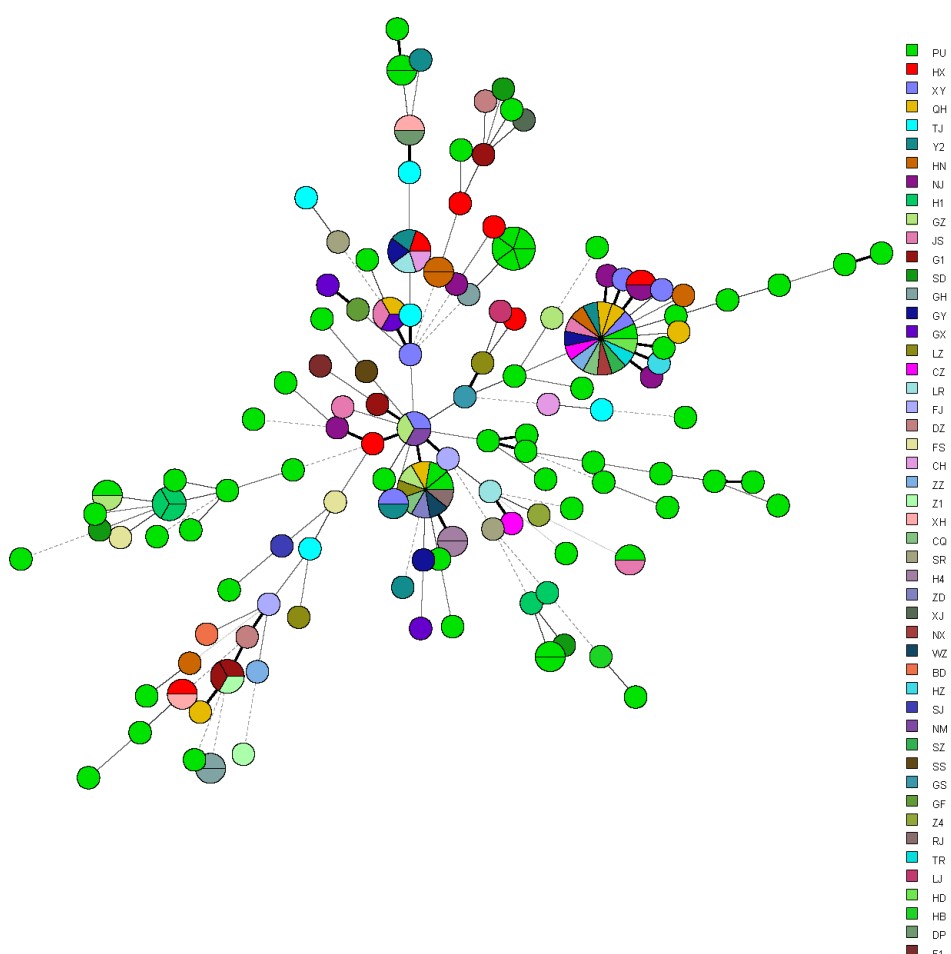

**FIG 2** Minimum spanning tree showing the distribution of the 145 genotypes (circles) found in the isolates studied and the size of the circle is correlated with the number of isolates. The connecting lines between the circles show the similarity between the profiles: thick solid lines indicate differences in only 1 marker, thin solid lines indicate differences in 2–3 marker and dashed lines denote differences in 4 or more markers.

from peripheral blood and catheter blood samples exhibited identical genotypes. For 2 patients who had 2 isolates cultured from peripheral blood collected at different dates (6 days interval and 18 days interval), their isolates exhibited identical genotypes. For the third patient, genotypes were different between the isolates from isolates cultured from peripheral blood collected 30 days interval. The genotypes differed in 7 markers. Two different genotypes were found in the patient with 3 isolates from peripheral blood, 1 causing the first episode and the other causing the second and third episode. The genotypes differed in only one marker.

## DISCUSSION

This study provides a comprehensive comparative analysis of *C. krusei* isolates obtained from bloodstream infections and other clinical specimen types, focusing on their biochemical characteristics, antifungal susceptibility profiles, and genotypic diversity as determined by microsatellite typing. Distinguishing true invasive vs superficial (or non-bloodstream) sources is challenging, and that some non-blood sites (e.g., ascites, wound exudates) may reflect secondary seeding. However, there is precedent in the literature for comparing blood vs non-blood/sterile non-blood *Candida* isolates to explore differences in antifungal susceptibility. For example, a study of *Candida glabrata* showed that isolates from the abdominal cavity (sterile non-blood site) had higher rates

**TABLE 4** Five isolates from five different patients' respiratory specimens shared identical genotype (Cluster 14, MT110)

| Patient no. | Gender | Age (year) | Date of admission | Specimen type | Specimen collection date | Isolate no. | No. of days post-admission with positive culture | Patient's bed No. | Diagnosis | Clinical outcomes |
|---|---|---|---|---|---|---|---|---|---|---|
| Patient 1 | M | 86 | 1/30/2023 | Tracheobronchial aspirates | 2/1/2023 | K143 | 2 | 2 | Severe pneumonia; COVID-19 infection (critical type); Type II respiratory failure; Hypertension (Grade 3, very high-risk group); Coronary atherosclerotic heart disease | Died after 38 days of hospitalization |
| Patient 2 | M | 37 | 2/14/2023 | Endotracheal aspirates | 2/19/2023 | K145 | 5 | 9 | Encephalitis; Status Epilepticus; Rhabdomyolysis; Bacteremia (Streptococcus mitis) | Transferred to another hospital after 8 days of hospitalization |
| Patient 3 | M | 76 | 3/1/2023 | Bronchoalveolar lavage fluid | 3/8/2023 | K147 | 7 | 11 | Lymphoma; Lung adenocarcinoma; Severe community acquired pneumonia (Viral pneumonia and respiratory failure caused by influenza A virus and COVID-19, combined with secondary bacterial infection); Type I respiratory failure; Septic shock; Coronary atherosclerotic heart disease; Acute coronary syndrome | Died after 47 days of hospitalization |
| Patient 4 | M | 83 | 5/19/2023 | Endotracheal aspirates | 6/1/2023 | K154 | 13 | 10 | Severe pneumonia; ARDS (heavy); COVID-19 infection (critical type); Type I respiratory failure; Sinus bradycardia; Hypertension (grade 3, very high-risk group) | Transferred to another hospital after 25 days of hospitalization |
| Patient 5 | M | 68 | 5/30/2023 | Tracheostomy secretions | 7/2/2023 | K157 | 33 | 9 | Squamous cell carcinoma of the right lung, treated with chemotherapy and immunosuppressive therapy for many times; Complicated with severe pneumonia and COVID-19 infection | Died after 51 days of hospitalization |

of resistance to fluconazole and echinocandins than blood culture isolates (13). Also, the ARTEMIS surveillance program reported that *C. krusei* isolates from urine (a non-blood specimen) had significantly lower susceptibility to voriconazole than blood isolates (14). In pediatric patients with *Candida* colonization of non-blood sites, *C. krusei* and other species showed reduced susceptibility compared to more sensitive species or baselines

(15). We believe the observed trends between bloodstream and non-blood isolates remain informative and clinically relevant and provide useful hypotheses for further work.

Interestingly, the prevalence of tyrosine arylamidase (TyrA) positivity was significantly higher among *C. krusei* isolates from non-blood specimens compared to bloodstream isolates. While the clinical implications of this finding remain unclear, it may suggest a potential link between TyrA activity and the pathogen's transition from colonization to invasive disease. This differential expression may reflect an adaptive metabolic shift favoring survival in the bloodstream, a nutrient-limited and immune-rich environment. Reduced TyrA activity could indicate a downregulation of tyrosine catabolism pathways, possibly to minimize immune detection or to adapt to the specific nutrient availability within the circulatory system. In other pathogenic fungi, such as *Penicillium marneffei*, tyrosine metabolism enzymes like HpdA have been shown to be critical for intracellular survival and yeast-phase growth, both of which are essential for pathogenesis (16). Similarly, in *Histoplasma capsulatum*, tyrosine catabolism contributes to the production of pyomelanin-like pigments, which are linked to virulence and resistance to oxidative stress (17). Pyomelanin derived from tyrosine degradation has also been implicated in enhancing the pathogenic potential of *Aspergillus fumigatus* by protecting against

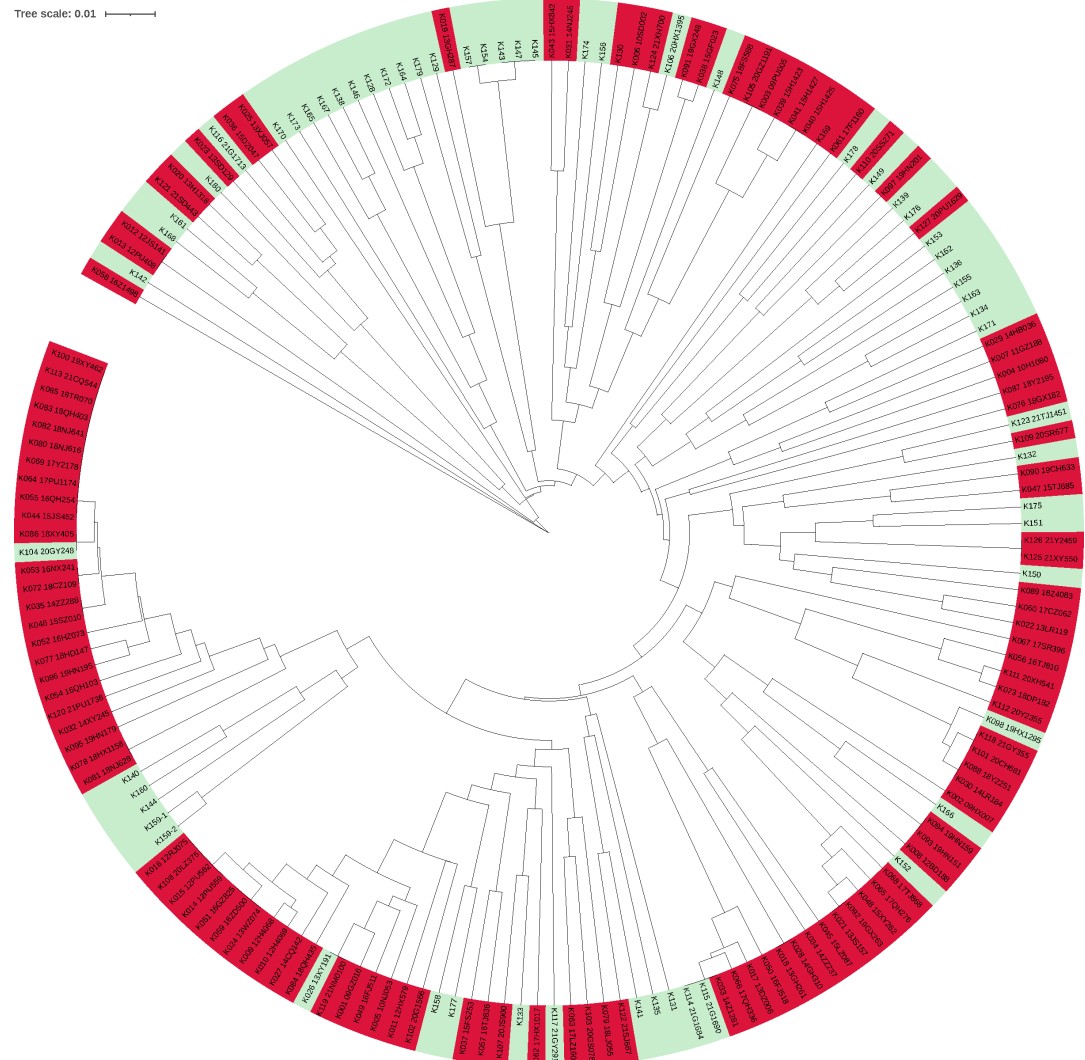

**FIG 3** Phylogenetic tree of *C. krusei* isolates based on eight microsatellite loci using MEGA (version 10.2.6) software and iTOL platform. Isolates are color-coded by specimen source: red = bloodstream, blue = non-blood. Outer rings indicate antifungal non-susceptibility profiles.

host immune defenses (18). While direct evidence in *C. krusei* is limited, these parallels suggest that tyrosine metabolism, and by extension TyrA activity, may play a role in the organism's ability to adapt to different host environments and stages of infection. Further research is warranted to elucidate the specific mechanisms by which TyrA activity influences *C. krusei* pathogenicity and its potential as a marker for invasive potential.

In terms of antifungal susceptibility, our study showed that bloodstream isolates exhibited lower minimum inhibitory concentrations (MICs) for several antifungal agents compared to non-blood isolates, most notably voriconazole and caspofungin. While *C. krusei* is intrinsically resistant to fluconazole, the observed variation in MIC values among isolates suggests heterogeneity in resistance levels, potentially influenced by prior antifungal exposure or selective pressure in hospital environments. This finding may reflect distinct selective pressures encountered by *C. krusei* in different host niches. Systemic antifungal exposure during candidemia can select for resistant strains in some settings; however, multiple lines of evidence indicate that non-blood (colonizing or device/sterile site) isolates may also, and under some circumstances more often, demonstrate reduced susceptibility compared with contemporaneous bloodstream isolates. First, colonizing isolates that have been exposed to prolonged azole therapy frequently show higher rates of azole nonsusceptibility than initial blood isolates, as documented in a multicenter study of post-treatment colonizing *Candida* isolates (19). Second, biofilm-associated and device-related isolates (which are more common among non-blood/sterile site specimens) display markedly increased antifungal tolerance compared with planktonic bloodstream isolates, a mechanism that can raise observed MICs and clinical resistance (20). Third, surveillance and comparative studies have reported higher MICs or lower azole susceptibility among non-blood specimens for certain species (including reports from large surveillance programs), supporting that source-dependent differences exist and are clinically relevant (13–15, 21).

Mechanistically, these observations can be explained by at least three, not-mutually-exclusive processes: (i) local/previous antifungal exposure at mucosal or device sites driving selection of resistant subpopulations before bloodstream invasion, (ii) biofilm formation on catheters or tissue surfaces that increases phenotypic tolerance to antifungals, and (iii) sampling/epidemiologic bias (different species distributions or patient populations at different sites). Reviews of *Candida* spp. resistance and acquired mechanisms summarize how prior antifungal exposure and adaptive genomic changes produce stable resistance in clinical isolates (22, 23). In addition, bloodstream invasion may require metabolic streamlining and loss of resistance-associated traits that carry a fitness cost in nutrient-limited, immune-challenged environments like the bloodstream. Previous studies have shown that antifungal resistance in *Candida* species can be associated with a fitness burden, including slower growth and reduced virulence *in vivo* (24–26). Therefore, bloodstream isolates may represent a subpopulation of *C. krusei* optimized for invasiveness rather than resistance. From a clinical perspective, these findings underscore the importance of site-specific susceptibility surveillance and support the idea that empiric antifungal therapy may be more effective against bloodstream *C. krusei* infections compared to superficial colonization or localized disease. Further genomic and transcriptomic profiling of isolates from different anatomical sites may elucidate the molecular basis of the observed differences in antifungal susceptibility.

In this study, there were some slight differences in MIC values between CLSI method and SYO. These discrepancies can be attributed to inherent methodological differences. CLSI relies on visual endpoint reading of growth inhibition in clear broth, while YeastOne incorporates a colorimetric indicator (alamarBlue) to quantify metabolic activity, which may detect growth inhibition earlier or more sensitively in some isolates. This can lead to slightly lower or higher MIC values depending on the drug-organism interaction (27, 28). In this study, while EA between the two methods is consistently high, CA varies depending on the specific antifungal agent. The relatively high rate of mE suggests that particular caution is needed when interpreting intermediate results. SYO is more

convenient for rapid and routine clinical use but given that CLSI serves as the reference (gold standard) method, particular attention should be paid when interpreting SYO results, especially in cases where MIC values are near clinical breakpoints or when minor error rates are elevated.

Microsatellite genotyping revealed considerable genetic diversity among the isolates but also highlighted the presence of certain genotypes that were predominantly found in bloodstream isolates. This raises the possibility of specific clonal lineages with enhanced pathogenic potential or hospital adaptation. The lack of identical genotypes among isolates from different patients argues against a common-source outbreak but supports the notion of strain-specific traits contributing to invasiveness. These observations are consistent with prior studies reporting genotype-specific variation in virulence and antifungal susceptibility in *C. krusei* and other non-*albicans Candida* species (4, 5, 29). In this study, we identified a genetically identical cluster of *C. krusei* isolates from five patients who were hospitalized concurrently in the same ward of a tertiary care hospital, they were all collected from critically ill patients with lower respiratory tract specimens (including tracheobronchial aspirates, endotracheal aspirates, and bronchoalveolar lavage) during a 5-month period were identified as belonging to the same microsatellite genotype. This clonal relatedness suggests the possible existence of a nosocomial transmission event or a shared environmental reservoir within the intensive care unit or associated healthcare environment. Despite differences in age, underlying conditions, and clinical outcomes, the genomic similarity among isolates points to a common origin. Clonal dissemination of *C. krusei* in hospital settings has been previously documented and is often associated with prolonged ICU stays, mechanical ventilation, and immunosuppression, all of which were present among the patients in this series (30–32).

Notably, among the five patients, four were diagnosed with severe pneumonia after COVID-19 infection. The overlap between COVID-19 infection and *C. krusei* colonization or infection is especially noteworthy. SARS-CoV-2 has been shown to significantly disrupt the respiratory mucosal barrier and alter the host immune landscape, predisposing patients to secondary fungal infections. In particular, the combined impact of immunosuppressive treatments (e.g., corticosteroids), prolonged ICU stays, mechanical ventilation, and invasive procedures likely facilitated fungal overgrowth and dissemination (33, 34). Previous studies have suggested that *C. krusei*, while inherently less virulent than other *Candida* spp., can act as an opportunistic pathogen under conditions of severe immunosuppression or microbiota imbalance. The high prevalence of the same genotype among patients with COVID-19 further supports the possibility that the SARS-CoV-2 pandemic may have amplified the risks of nosocomial colonization and outbreaks by certain fungal clones. These findings highlight the need for enhanced fungal surveillance, particularly in ICU settings during viral epidemics.

This finding highlights the critical importance of molecular epidemiological surveillance in detecting cryptic clusters of healthcare-associated fungal infections. Genotyping tools such as microsatellite typing can uncover hidden outbreaks that might otherwise be missed based on phenotypic identification alone. The clustering of genetically identical strains in the respiratory tract also raises questions about *C. krusei*'s capacity for colonization vs true infection in the setting of severe viral pneumonia or COVID-19, and further study is warranted to clarify its pathogenic role under these conditions.

Together, these data suggest that a subset of *C. krusei* strains may possess a combination of metabolic, genetic, and resistance-related features that enhance their ability to cause bloodstream infections. Although this study does not establish a direct causal relationship between specific traits and invasiveness, the consistent patterns observed provide a valuable starting point for future investigations. Elucidating the molecular mechanisms underlying these associations—such as through transcriptomic profiling, virulence assays, or host-pathogen interaction models—could reveal novel targets for risk prediction or therapeutic intervention. This study has several limitations. First, the

sample size was limited, which may restrict the generalizability of the findings. Second, the study was observational in nature and did not include functional validation of putative invasiveness-associated traits. Finally, because it was a retrospective analysis of collected isolates, detailed clinical information such as prior antifungal treatment, underlying diseases and patient outcomes, was not systematically available and incomplete. This limited our ability to determine whether resistance to certain antifungal agents was intrinsic or developed during antifungal exposure. Future prospective studies with more complete clinical data are needed to further clarify these issues.

In conclusion, this study highlights important biochemical, antifungal susceptibility, and genotypic differences between *C. krusei* isolates from bloodstream and non-blood clinical samples. These differences may serve as indicators of invasive potential and warrant further investigation to better understand *C. krusei* pathogenesis and guide clinical management of infections caused by this emerging fungal pathogen.

## ACKNOWLEDGMENTS

We really appreciate all participants of the CHIF-NET study. This work was supported by the Natural Science Foundation of China (Grant number: 82202592) and the National High Level Hospital Clinical Research Funding (Grant number: 2022-PUMCH-C-052).

Ying Zhao and Yingchun Xu: Conceptualization. Ying Zhao: Data curation; formal analysis; software; visualization; writing—original draft. Ying Zhao, Lina Guo, and Yingchun Xu: Funding acquisition. Han Wang, JinhanYu, Yi Li, Lina Guo, Ge Zhang, Wei Kang, Meng Xiao, and Qiwen Yang: Methodology. Ying Zhao: Project administration. Meng Xiao, Qiwen Yang, Lina Guo, and Yingchun Xu: Resources. Lina Guo, Yingchun Xu and Ying Zhao: Writing—review and editing.

## AUTHOR AFFILIATIONS

[1]Department of Laboratory Medicine, State Key Laboratory of Complex, Severe, and Rare Diseases, Peking Union Medical College Hospital, Chinese Academy of Medical Sciences and Peking Union Medical College, Beijing, China
[2]Graduate School, Chinese Academy of Medical Sciences and Peking Union Medical College, Beijing, China

## AUTHOR ORCIDs

Ying Zhao http://orcid.org/0000-0002-7093-1121
Jinhan Yu http://orcid.org/0000-0003-3807-2088
Meng Xiao http://orcid.org/0000-0003-2103-7008
Qiwen Yang http://orcid.org/0000-0001-7272-3900
Lina Guo http://orcid.org/0000-0003-4812-8663
Yingchun Xu http://orcid.org/0000-0002-7126-9459

## FUNDING

| Funder | Grant(s) | Author(s) |
| --- | --- | --- |
| National High Level Hospital Clinical Research Funding | 2022-PUMCH-C-052 | Ying Zhao |
| Natural Science Foundation of Henan Province | 82202592 | Ying Zhao |

## AUTHOR CONTRIBUTIONS

Ying Zhao, Conceptualization, Data curation, Formal analysis, Investigation, Methodology, Project administration, Resources, Software, Visualization | Han Wang, Methodology | Jinhan Yu, Methodology | Yi Li, Methodology | Ge Zhang, Methodology | Wei Kang, Methodology | Meng Xiao, Methodology, Resources | Qiwen Yang, Methodology,

Resources, Supervision | Lina Guo, Methodology, Resources | Yingchun Xu, Conceptualization, Methodology, Resources, Supervision

## DATA AVAILABILITY

The authors confirm that the data supporting the findings of this study are available within the article.

## ADDITIONAL FILES

The following material is available online.

### Supplemental Material

**Table S1 (Spectrum02608-25-S0001.xlsx).** Clinical information of isolates source.
**Table S2 (Spectrum02608-25-S0002.xlsx).** Designation of 120 MT types generated by microsatellite typing.

### Open Peer Review

**PEER REVIEW HISTORY (review-history.pdf).** An accounting of the reviewer comments and feedback.

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
