## [Reviewer comments · Microbiology Spectrum]

Microbiology Spectrum

Phenotypic and Genotypic Differences Between Bloodstream and Non-Blood *Candida krusei* Isolates: Implications for Invasiveness and Antifungal Susceptibility

Ying Zhao, Han Wang, Jinhan Yu, Yi Li, Ge Zhang, Wei Kang, Meng Xiao, Qiwen Yang, Li-Na Guo, and Ying-Chun Xu

Corresponding Author(s): Ying Zhao, Peking Union Medical College Hospital

Review Timeline:

Submission Date:	August 21, 2025
Editorial Decision:	September 26, 2025
Revision Received:	October 9, 2025
Editorial Decision:	November 5, 2025
Revision Received:	November 5, 2025
Accepted:	November 10, 2025

Editor: Matthew Anderson

Reviewer(s): Disclosure of reviewer identity is with reference to reviewer comments included in decision letter(s). The following individuals involved in review of your submission have agreed to reveal their identity: Ehab A. Salama (Reviewer #1)

Transaction Report:

DOI: <https://doi.org/10.1128/spectrum.02608-25>

Re: Spectrum02608-25 (Phenotypic and Genotypic Differences Between Bloodstream and Non-Blood Candida krusei Isolates: Implications for Invasiveness and Antifungal Susceptibility)

Dear Prof. Ying Zhao:

Thank you for the privilege of reviewing your work. Below you will find my comments, instructions from the Spectrum editorial office, and the reviewer comments.

Please note Reviewer 1's comments on the distinction of bloodstream and non-bloodstream isolates. This will be an important distinction to make in the revision. Also consider editing by someone using English as a first language or an English-language editing service to address concerns with grammar.

Revision Guidelines

Sincerely,
Matthew Anderson
Editor
Microbiology Spectrum

Reviewer #1 (Comments for the Author):

The manuscript addresses a clinically important topic on the biochemical and genetic variation in *Candida krusei* isolates with potential medical impact. However, the presentation of results is not clear and does not support robust or reliable conclusions.

Major Critiques

The study does not convincingly differentiate between bloodstream and non-blood *Candida krusei* isolates. Several of the so-called "non-blood" samples (e.g., respiratory specimens, bronchoalveolar lavage, ascitic fluid, wound secretions) could easily represent secondary seeding from bloodstream infection. This distinction is particularly problematic since the manuscript focuses on invasiveness. A valid comparison would require categorizing isolates from blood versus a truly superficial site, such as vaginal samples.

The supplementary table presents genetic diversity data only for isolates 1-7. The supplementary material should comprehensively include genetic diversity results for all isolates to allow proper evaluation.

The number of isolates analyzed is very limited, which weakens the generalizability of the findings.

Minor Critiques

Materials and methods: Reference 5 does not address microsatellite markers or genotyping and is therefore inappropriate in this context.

MIC Analysis: Differences between the CLSI method and SYO for MIC determination are not central to the study's objectives and should be removed.

Line 59: The word "paediatric" should be corrected to "Pediatric"

Line 104-110: description of the PUMCH hospital which is not the scope of the study and should be removed.

Line 159: The word "denaturationa" should be corrected to "denaturation"

Line 185: The word "ematologic" should be replaced with "hematologic"

Table 1 containing data for only 6 patients who have multiple isolates

Table 2 use the word "Antibiotic" should be replaced with "Antifungal"

Table 2: Remove the agreement/error data, which complicates interpretation without adding significant value.

Reviewer #2 (Comments for the Author):

COMMENTS TO AUTHORS

Lines 30, 90, 102: *C. krusei* must be written in italics.

Line 43: The Fungal name is misspelled; it must be corrected

Lines 58, 61: We recommend that the authors write the whole name of the species at the beginning of the sentences.

Line 113, 119, 120, 123: The city name is missing from the Biomerieux reference.

Line 133: Change *C. parapsilosis* to *Candida parapsilosis* and *Candida krusei* to *C. krusei*.

Line 154: If you developed the DNA extraction protocol, a detailed description of it should be provided. If it is based on a protocol by other authors, you should include the citation.

Line 166: The city name is missing from the ABI reference.

Line 185: "hematologic" is misspelled; it must be corrected.

Line 187: We highly recommend that you erase "etc." and provide details of those 25 patients. Why did you decide to detail some of the clinical conditions and not all of them?

Line 241: Write "in blood samples, it WAS 0 %" instead of "in blood samples, it IS 0 %".

Lines 319-321: We somehow disagree with this conclusion made by Ying Zhao and colleagues: "Bloodstream isolates are more likely to be subject to systemic antifungal therapy, particularly in hospitalized patients with invasive candidiasis, which may selectively FAVOR STRAINS THAT RETAIN SUSCEPTIBILITY". Isolates that have been exposed to antifungals and have survived in their presence tend to develop resistance, at least in *Candida albicans* isolates. Therefore, we would expect higher MIC values for bloodstream isolates rather than non-blood isolates. How can you explain this? Could you provide any reference to support your conclusion?

Line 328: *Candida* must be written in italics.

Line 382: spp. must not be written in italics.

Lines 408-410: It should be very interesting to know this clinical data; this would make it possible to determine whether the resistance to some antifungal compounds is intrinsic or has been developed during antifungal exposure.

Line 419: "Susceptibility" is misspelled; it must be corrected.

Line 452: The names of all microorganisms in references must be written in italics.

Revision

The manuscript addresses a clinically important topic on the biochemical and genetic variation in *Candida krusei* isolates with potential medical impact. However, the presentation of results is not clear and does not support robust or reliable conclusions.

Major Critiques

The study does not convincingly differentiate between bloodstream and non-blood *Candida krusei* isolates. Several of the so-called “non-blood” samples (e.g., respiratory specimens, bronchoalveolar lavage, ascitic fluid, wound secretions) could easily represent secondary seeding from bloodstream infection. This distinction is particularly problematic since the manuscript focuses on invasiveness. A valid comparison would require categorizing isolates from blood versus a truly superficial site, such as vaginal samples.

The supplementary table presents genetic diversity data only for isolates 1–7. The supplementary material should comprehensively include genetic diversity results for all isolates to allow proper evaluation.

The number of isolates analyzed is very limited, which weakens the generalizability of the findings.

Minor Critiques

Materials and methods: Reference 5 does not address microsatellite markers or genotyping and is therefore inappropriate in this context.

MIC Analysis: Differences between the CLSI method and SYO for MIC determination are not central to the study’s objectives and should be removed.

Line 59: The word “paediatric” should be corrected to “Pediatric”

Line 104-110: description of the PUMCH hospital which is not the scope of the study and should be removed.

Line 159: The word “denaturational” should be corrected to “denaturation”

Line 185: The word “ematologic” should be replaced with “hematologic”

Table 1 containing data for only 6 patients who have multiple isolates

Table 2 use the word "Antibiotic" should be replaced with “Antifungal”

Table 2: Remove the agreement/error data, which complicates interpretation without adding significant value.

COMMENTS TO EDITOR:

Ying Zhao and colleagues' research is focused on the characterization of several *Candida krusei* clinal isolates, which is an emerging fungal pathogen; not only that, *C. krusei* infection poses a big challenge for physicians due to its intrinsic resistance to fluconazole. One of the main strengths of this investigation is that the authors have analysed many isolates. Although we have a few comments and corrections, we believe this paper must be published in the "Microbiology Spectrum" journal.

COMMENTS TO AUTHORS

Lines 30, 90, 102: *C. krusei* must be written in italics.

Line 43: The Fungal name is misspelled; it must be corrected

Lines 58, 61: We recommend that the authors write the whole name of the species at the beginning of the sentences.

Line 113, 119, 120, 123: The city name is missing from the Biomerieux reference.

Line 133: Change *C. parapsilosis* to *Candida parapsilosis* and *Candida krusei* to *C. krusei*.

Line 154: If you developed the DNA extraction protocol, a detailed description of it should be provided. If it is based on a protocol by other authors, you should include the citation.

Line 166: The city name is missing from the ABI reference.

Line 185: "hematologic" is misspelled; it must be corrected.

Line 187: We highly recommend that you erase "etc." and provide details of those 25 patients. Why did you decide to detail some of the clinical conditions and not all of them?

Line 241: Write "in blood samples, it WAS 0 %" instead of "in blood samples, it IS 0 %".

Lines 319-321: We somehow disagree with this conclusion made by Ying Zhao and colleagues: "Bloodstream isolates are more likely to be subject to systemic antifungal therapy, particularly in hospitalized patients with invasive candidiasis, which may selectively FAVOR STRAINS THAT RETAIN SUSCEPTIBILITY". Isolates that have been exposed to antifungals and have survived in their presence tend to develop resistance, at least in *Candida albicans* isolates. Therefore, we would expect higher MIC values for bloodstream isolates rather than non-blood isolates. How can you explain this? Could you provide any reference to support your conclusion?

Line 328: *Candida* must be written in italics.

Line 382: spp. must not be written in italics.

Lines 408-410: It should be very interesting to know this clinical data; this would make it possible to determine whether the resistance to some antifungal compounds is intrinsic or has been developed during antifungal exposure.

Line 419: "Susceptibility" is misspelled; it must be corrected.

Line 452: The names of all microorganisms in references must be written in italics.

Reviewer #1:

The manuscript addresses a clinically important topic on the biochemical and genetic variation in *Candida krusei* isolates with potential medical impact. However, the presentation of results is not clear and does not support robust or reliable conclusions.

Major Critiques

The study does not convincingly differentiate between bloodstream and non-blood *Candida krusei* isolates. Several of the so-called "non-blood" samples (e.g., respiratory specimens, bronchoalveolar lavage, ascitic fluid, wound secretions) could easily represent secondary seeding from bloodstream infection. This distinction is particularly problematic since the manuscript focuses on invasiveness. A valid comparison would require categorizing isolates from blood versus a truly superficial site, such as vaginal samples.

We agree that distinguishing true invasive vs superficial (or non-bloodstream) sources is challenging, and that some non-blood sites (e.g. respiratory secretions, ascites, wound exudates) may reflect secondary seeding. However, there is precedent in the literature for comparing blood vs non-blood/sterile non-blood Candida isolates to explore differences in antifungal susceptibility. For example, a study of Candida glabrata showed that isolates from the abdominal cavity (sterile non-blood site) had higher rates of resistance to fluconazole and echinocandins than blood culture isolates [1]. Also, the ARTEMIS surveillance program reported that C. krusei isolates from urine (a non-blood specimen) had significantly lower susceptibility to voriconazole than blood isolates [2]. In pediatric patients with Candida colonization of non-blood sites, C. krusei and other species showed reduced susceptibility compared to more sensitive species or baselines [3]. We have now clarified this point in the Methods section and acknowledged this limitation in the Discussion. Despite this, we believe the observed trends between bloodstream and non-blood isolates remain informative and clinically relevant and provide useful hypotheses for further work.

Reference

[1] Díaz-García J, Mesquida A, Gómez A, Machado M, Martín-Rabadán P, Alcalá L, Sánchez-Carrillo C, Reigadas E, Vicente T, Muñoz P, Escribano P, Guinea J. Antifungal Susceptibility Testing Identifies the Abdominal Cavity as a Source of *Candida glabrata*-Resistant Isolates. *Antimicrob Agents Chemother.* 2021 Nov 17;65(12):e0124921.

[2] Pfaller MA, Diekema DJ, Gibbs DL, Newell VA, Nagy E, Dobiasova S, Rinaldi M, Barton R, Veselov A; Global Antifungal Surveillance Group. *Candida krusei*, a multidrug-resistant opportunistic fungal pathogen: geographic and temporal trends from the ARTEMIS DISK Antifungal Surveillance Program, 2001 to 2005. *J Clin Microbiol.* 2008 Feb;46(2):515-21.

[3] Badiie P, Choopanizadeh M, Moghadam AG, Nasab AH, Jafarian H, Shamsizadeh A, Soltani J. Antifungal susceptibility patterns of colonized *Candida* species isolates from immunocompromised pediatric patients in five university hospitals. *Iran J Microbiol.* 2017 Dec;9(6):363-371.

The supplementary table presents genetic diversity data only for isolates 1-7. The supplementary material should comprehensively include genetic diversity results for all isolates to allow proper evaluation.

We provide the complete isolates data in the supplementary materials (Supplement Table S1) .

The number of isolates analyzed is very limited, which weakens the generalizability of the findings.

We appreciate the reviewer's comment and agree that the number of isolates analyzed is limited, which may restrict the generalizability of the findings. We emphasized this in our discussion (clean version Line 431-432). However, C. krusei invasive infections are relatively uncommon, and our study included a comparatively large collection of isolates analyzed. Despite the limited sample size, the results provide valuable insights into the phenotypic and genotypic characteristics of clinical C. krusei isolates and may serve as a useful reference for future multicenter studies.

Minor Critiques

Materials and methods: Reference 5 does not address microsatellite markers or genotyping and is therefore inappropriate in this context.

We thank the reviewer for the careful reading. The citation was incorrectly numbered in the text. The correct reference should be Reference [7], which addresses microsatellite markers and genotyping. This error has now been corrected in the revised manuscript.

MIC Analysis: Differences between the CLSI method and SYO for MIC determination are not central to the study's objectives and should be removed.

*We appreciate the reviewer's valuable comment. We agree that the main objective of our study is not a direct comparison between CLSI and SYO methods. However, we believe it is clinically relevant to present these data. The CLSI broth microdilution method is the reference standard, but it is technically demanding and rarely used in routine clinical laboratories. In contrast, the Sensititre YeastOne (SYO) commercial method is widely used in clinical practice due to its convenience. Highlighting the discrepancies observed between these two methods in *C. krusei* testing may help clinicians better interpret MIC results and make more informed treatment decisions. Previous studies have reported discrepancies between the two methods in antifungal susceptibility testing, but the number of *Candida auris* isolates included was limited [1-3]. Our study included a relatively large number of *C. krusei* isolates tested by both methods, which provides valuable reference data. Therefore, we believe this information adds significance and have retained it in the manuscript. Therefore, we believe this information adds significance and have retained it in the manuscript.*

*[1] Altınbaş R, Barış A, Şen S, Öztürk R, Kiraz N. Comparison of the Sensititre YeastOne antifungal method with the CLSI M27-A3 reference method to determine the activity of antifungal agents against clinical isolates of *Candida* spp. Turk J Med Sci. 2020 Dec 17;50(8):2024-2031.*

*[2] Lim HJ, Shin JH, Kim MN, Yong D, Byun SA, Choi MJ, Lee SY, Won EJ, Kee SJ, Kim SH, Shin MG. Evaluation of Two Commercial Broth Microdilution Methods Using Different Interpretive Criteria for the Detection of Molecular Mechanisms of Acquired Azole and Echinocandin Resistance in Four Common *Candida* Species. Antimicrob Agents Chemother. 2020 Oct 20;64(11):e00740-20.*

[3] Ceballos-Garzon A, Holzapfel M, Welsch J, Mercer D. Identification and antifungal susceptibility patterns of reference yeast strains to novel and conventional agents: a comparative study using CLSI, EUCAST and Sensititre YeastOne methods. JAC Antimicrob Resist. 2025 Mar 19;7(2):dlaf040.

Line 59: The word "paediatric" should be corrected to "Pediatric"

The word has been corrected from "paediatric" to "pediatric" in the revised manuscript.

Line 104-110: description of the PUMCH hospital which is not the scope of the study and should be removed.

We appreciate the reviewer's suggestion. We agree that a lengthy description of PUMCH is beyond the scope of this study. Accordingly, we have shortened the section and retained only essential information to clarify the source of isolates.

Line 159: The word "denaturiona" should be corrected to "denaturation"

The word has been corrected from "denaturiona" to "denaturation" in the revised manuscript.

Line 185: The word "ematologic" should be replaced with "hematologic"

The word has been corrected from "ematologic" to "hematologic" in the revised manuscript.

Table 1 containing data for only 6 patients who have multiple isolates

Table 1 presents data only for the six patients who had multiple C. krusei isolates, to summarize cases with repeated isolates for clarity. We provide the complete isolates data in the supplementary materials (Supplement Table S1) .

Table 2 use the word "Antibiotic" should be replaced with "Antifungal"

The word has been corrected from "Antibiotic" to "Antifungal" in the revised manuscript.

Table 2: Remove the agreement/error data, which complicates interpretation without adding significant value.

While we understand that including agreement/error data may complicate interpretation, we believe that presenting the differences between CLSI and SYO MIC results is clinically valuable. Retaining this information helps clinicians better interpret antifungal susceptibility results and guide therapy, which we consider important for the manuscript.

Reviewer #2 (Comments for the Author):

COMMENTS TO AUTHORS

Lines 30, 90, 102: C. krusei must be written in italics.

We thank the reviewer for noticing this. All instances of C. krusei have been corrected to italics in the revised manuscript.

Line 43: The Fungal name is misspelled; it must be corrected

The fungal name has been corrected in the revised manuscript.

Lines 58, 61: We recommend that the authors write the whole name of the species at the beginning of the sentences.

We thank the reviewer for the suggestion. The full species name is already provided at the beginning of the paragraph; for readability and brevity, we have used the abbreviated form (C. krusei) in subsequent mentions.

Line 113, 119, 120, 123: The city name is missing from the Biomerieux reference.

The city name has been added to the Biomerieux references in the revised manuscript.

Line 133: Change C. parapsilosis to Candida parapsilosis and Candida krusei to C. krusei.

The species names have been corrected to Candida parapsilosis and C. krusei in the revised manuscript.

Line 154: If you developed the DNA extraction protocol, a detailed description of it should be provided. If it is based on a protocol by other authors, you should include the citation.

The genomic DNA was extracted using the Fungi Genomic DNA Extraction Kit (Solarbio Science & Technology, Beijing, China) according to the company's recommended protocols. We have revised the manuscript accordingly.

Line 166: The city name is missing from the ABI reference.

The city name has been added to the ABI references in the revised manuscript.

Line 185: "hematologic" is misspelled; it must be corrected.

The word has been corrected from "ematologic" to "hematologic" in the revised manuscript.

Line 187: We highly recommend that you erase "etc." and provide details of those 25 patients. Why did you decide to detail some of the clinical conditions and not all of them?

We provide the complete patient data in the supplementary materials (Supplement Table S1).

Line 241: Write "in blood samples, it WAS 0 %" instead of "in blood samples, it IS 0 %".

We have revised the sentence as suggested and changed "it is 0%" to "it was 0%".

Lines 319-321: We somehow disagree with this conclusion made by Ying Zhao and colleagues: "Bloodstream isolates are more likely to be subject to systemic antifungal therapy, particularly in hospitalized patients with invasive candidiasis, which may selectively FAVOR STRAINS THAT RETAIN SUSCEPTIBILITY". Isolates that have been exposed to antifungals and have survived in their presence tend to develop resistance, at least in *Candida albicans* isolates. Therefore, we would expect higher MIC values for bloodstream isolates rather than non-blood isolates. How can you explain this? Could you provide any reference to support your conclusion?

*We thank the reviewer for this important point. We agree that systemic antifungal exposure during candidemia can select for resistant strains in some settings; however, multiple lines of evidence indicate that non-blood (colonizing or device/sterile-site) isolates may also — and under some circumstances more often — demonstrate reduced susceptibility compared with contemporaneous bloodstream isolates. First, colonizing isolates that have been exposed to prolonged azole therapy frequently show higher rates of azole nonsusceptibility than initial blood isolates, as documented in a multicenter study of post-treatment colonizing *Candida* isolates [1]. Second, biofilm-associated and device-related isolates (which are more common among non-blood/sterile site specimens) display markedly increased antifungal tolerance compared with planktonic bloodstream isolates, a mechanism that can raise observed MICs and clinical resistance [2]. Third, surveillance and comparative studies have reported higher MICs or lower azole susceptibility among non-blood specimens for certain species (including reports from large surveillance programs), supporting that source-dependent differences exist and are clinically relevant [3-6].*

*Mechanistically, these observations can be explained by at least three, not-mutually-exclusive processes: (1) local/prior antifungal exposure at mucosal or device sites driving selection of resistant subpopulations before bloodstream invasion; (2) biofilm formation on catheters or tissue surfaces that increases phenotypic tolerance to antifungals; and (3) sampling/epidemiologic bias (different species distributions or patient populations at different sites). Reviews of *Candida* resistance and acquired mechanisms summarize how prior drug exposure and adaptive genomic changes produce stable resistance in clinical isolates [7-8].*

In view of these points, we believe the expectation that bloodstream isolates must always have higher MICs is not universally supported by the literature. Nevertheless, we acknowledge that both possibilities exist and that distinguishing true primary colonization from secondary seeding remains a limitation in single-center series. We have therefore revised the Discussion to (1) cite the above literature, (2) explain the plausible mechanisms for higher MICs in non-blood isolates (prior local exposure, biofilms, species mix), and (3) explicitly acknowledge the sampling-bias limitation and the need for prospective, paired (site + blood) studies to resolve this question definitively.

*[1] Jensen RH, Johansen HK, Søres LM, Lemming LE, Rosenvinge FS, Nielsen L, Olesen B, Kristensen L, Dzajic E, Astvad KM, Arendrup MC. Posttreatment Antifungal Resistance among Colonizing *Candida* Isolates in Candidemia Patients: Results from a Systematic Multicenter Study. *Antimicrob Agents Chemother.* 2015 Dec 28;60(3):1500-8. doi: 10.1128/AAC.01763-15.*

*[2] Silva S, Rodrigues CF, Araújo D, Rodrigues ME, Henriques M. *Candida* Species Biofilms' Antifungal Resistance. *J Fungi (Basel).* 2017 Feb 21;3(1):8.*

[3] Díaz-García J, Mesquida A, Gómez A, Machado M, Martín-Rabadán P, Alcalá L, Sánchez-Carrillo C, Reigadas E,

Vicente T, Muñoz P, Escribano P, Guinea J. Antifungal Susceptibility Testing Identifies the Abdominal Cavity as a Source of *Candida glabrata*-Resistant Isolates. *Antimicrob Agents Chemother*. 2021 Nov 17;65(12):e0124921.

[4] Pfaller MA, Diekema DJ, Gibbs DL, Newell VA, Nagy E, Dobiasova S, Rinaldi M, Barton R, Veselov A; Global Antifungal Surveillance Group. *Candida krusei*, a multidrug-resistant opportunistic fungal pathogen: geographic and temporal trends from the ARTEMIS DISK Antifungal Surveillance Program, 2001 to 2005. *J Clin Microbiol*. 2008 Feb;46(2):515-21.

[5] Badiie P, Choopanizadeh M, Moghadam AG, Nasab AH, Jafarian H, Shamsizadeh A, Soltani J. Antifungal susceptibility patterns of colonized *Candida* species isolates from immunocompromised pediatric patients in five university hospitals. *Iran J Microbiol*. 2017 Dec;9(6):363-371.

[6] Singh R, Kaur M, Chakrabarti A, Shankarnarayan SA, Rudramurthy SM. Biofilm formation by *Candida auris* isolated from colonising sites and candidemia cases. *Mycoses*. 2019 Aug;62(8):706-709.

[7] Bhattacharya S, Sae-Tia S, Fries BC. Candidiasis and Mechanisms of Antifungal Resistance. *Antibiotics (Basel)*. 2020 Jun 9;9(6):312.

[8] Morais Vasconcelos Oliveira J, Conceição Oliver J, Latércia Tranches Dias A, Barbosa Padovan AC, Siqueira Caixeta E, Caixeta Franco Ariosa M. Detection of ERG11 Overexpression in *Candida albicans* isolates from environmental sources and clinical isolates treated with inhibitory and subinhibitory concentrations of fluconazole. *Mycoses*. 2021 Feb;64(2):220-227.

Line 328: *Candida* must be written in italics.

It has been corrected to italics in the revised manuscript.

Line 382: spp. must not be written in italics.

It has been corrected in the revised manuscript.

Lines 408-410: It should be very interesting to know this clinical data; this would make it possible to determine whether the resistance to some antifungal compounds is intrinsic or has been developed during antifungal exposure.

Thank you very much for this insightful comment. We fully agree that clinical data such as prior antifungal treatment, underlying diseases, and patient outcomes would provide valuable context for interpreting strain behavior. However, as this was a retrospective study based on the collection of isolates and available information, these clinical data were not systematically recorded and are therefore incomplete. We have clarified this limitation in the revised manuscript.

Line 419: "Susceptibility" is misspelled; it must be corrected.

We have corrected the word in the revised manuscript.

Line 452: The names of all microorganisms in references must be written in italics.

We have corrected all microorganisms' names in references in the revised manuscript.

Re: Spectrum02608-25R1 (Phenotypic and Genotypic Differences Between Bloodstream and Non-Blood *Candida krusei* Isolates: Implications for Invasiveness and Antifungal Susceptibility)

Dear Prof. Ying Zhao:

Thank you for the privilege of reviewing your work. Below you will find my comments, instructions from the Spectrum editorial office, and the reviewer comments.

There are very minor adjustments needed to the revised version of the manuscript. After making these small text changes, please submit the revised manuscript.

Revision Guidelines

Sincerely,
Matthew Anderson
Editor
Microbiology Spectrum

Reviewer #1 (Comments for the Author):

The authors have adequately addressed all the points raised in the previous revision. A few additional minor points should be corrected as follows:

- Table 1: Define the abbreviation FUO.

- Table 3: In the title, change "14 clusters found in microsatellite typing" to "Fourteen clusters found in microsatellite typing."

Reviewer #1:

The manuscript addresses a clinically important topic on the biochemical and genetic variation in *Candida krusei* isolates with potential medical impact. However, the presentation of results is not clear and does not support robust or reliable conclusions.

Major Critiques

The study does not convincingly differentiate between bloodstream and non-blood *Candida krusei* isolates. Several of the so-called "non-blood" samples (e.g., respiratory specimens, bronchoalveolar lavage, ascitic fluid, wound secretions) could easily represent secondary seeding from bloodstream infection. This distinction is particularly problematic since the manuscript focuses on invasiveness. A valid comparison would require categorizing isolates from blood versus a truly superficial site, such as vaginal samples.

We agree that distinguishing true invasive vs superficial (or non-bloodstream) sources is challenging, and that some non-blood sites (e.g. respiratory secretions, ascites, wound exudates) may reflect secondary seeding. However, there is precedent in the literature for comparing blood vs non-blood/sterile non-blood Candida isolates to explore differences in antifungal susceptibility. For example, a study of Candida glabrata showed that isolates from the abdominal cavity (sterile non-blood site) had higher rates of resistance to fluconazole and echinocandins than blood culture isolates [1]. Also, the ARTEMIS surveillance program reported that C. krusei isolates from urine (a non-blood specimen) had significantly lower susceptibility to voriconazole than blood isolates [2]. In pediatric patients with Candida colonization of non-blood sites, C. krusei and other species showed reduced susceptibility compared to more sensitive species or baselines [3]. We have now clarified this point in the Methods section and acknowledged this limitation in the Discussion. Despite this, we believe the observed trends between bloodstream and non-blood isolates remain informative and clinically relevant and provide useful hypotheses for further work.

Reference

[1] Díaz-García J, Mesquida A, Gómez A, Machado M, Martín-Rabadán P, Alcalá L, Sánchez-Carrillo C, Reigadas E, Vicente T, Muñoz P, Escribano P, Guinea J. Antifungal Susceptibility Testing Identifies the Abdominal Cavity as a Source of *Candida glabrata*-Resistant Isolates. *Antimicrob Agents Chemother.* 2021 Nov 17;65(12):e0124921.

[2] Pfaller MA, Diekema DJ, Gibbs DL, Newell VA, Nagy E, Dobiasova S, Rinaldi M, Barton R, Veselov A; Global Antifungal Surveillance Group. *Candida krusei*, a multidrug-resistant opportunistic fungal pathogen: geographic and temporal trends from the ARTEMIS DISK Antifungal Surveillance Program, 2001 to 2005. *J Clin Microbiol.* 2008 Feb;46(2):515-21.

[3] Badiie P, Choopanizadeh M, Moghadam AG, Nasab AH, Jafarian H, Shamsizadeh A, Soltani J. Antifungal susceptibility patterns of colonized *Candida* species isolates from immunocompromised pediatric patients in five university hospitals. *Iran J Microbiol.* 2017 Dec;9(6):363-371.

The supplementary table presents genetic diversity data only for isolates 1-7. The supplementary material should comprehensively include genetic diversity results for all isolates to allow proper evaluation.

We provide the complete isolates data in the supplementary materials (Supplement Table S1) .

The number of isolates analyzed is very limited, which weakens the generalizability of the findings.

We appreciate the reviewer's comment and agree that the number of isolates analyzed is limited, which may restrict the generalizability of the findings. We emphasized this in our discussion (clean version Line 431-432). However, C. krusei invasive infections are relatively uncommon, and our study included a comparatively large collection of isolates analyzed. Despite the limited sample size, the results provide valuable insights into the phenotypic and genotypic characteristics of clinical C. krusei isolates and may serve as a useful reference for future multicenter studies.

Minor Critiques

Materials and methods: Reference 5 does not address microsatellite markers or genotyping and is therefore inappropriate in this context.

We thank the reviewer for the careful reading. The citation was incorrectly numbered in the text. The correct reference should be Reference [7], which addresses microsatellite markers and genotyping. This error has now been corrected in the revised manuscript.

MIC Analysis: Differences between the CLSI method and SYO for MIC determination are not central to the study's objectives and should be removed.

*We appreciate the reviewer's valuable comment. We agree that the main objective of our study is not a direct comparison between CLSI and SYO methods. However, we believe it is clinically relevant to present these data. The CLSI broth microdilution method is the reference standard, but it is technically demanding and rarely used in routine clinical laboratories. In contrast, the Sensititre YeastOne (SYO) commercial method is widely used in clinical practice due to its convenience. Highlighting the discrepancies observed between these two methods in *C. krusei* testing may help clinicians better interpret MIC results and make more informed treatment decisions. Previous studies have reported discrepancies between the two methods in antifungal susceptibility testing, but the number of *Candida auris* isolates included was limited [1-3]. Our study included a relatively large number of *C. krusei* isolates tested by both methods, which provides valuable reference data. Therefore, we believe this information adds significance and have retained it in the manuscript. Therefore, we believe this information adds significance and have retained it in the manuscript.*

*[1] Altınbaş R, Barış A, Şen S, Öztürk R, Kiraz N. Comparison of the Sensititre YeastOne antifungal method with the CLSI M27-A3 reference method to determine the activity of antifungal agents against clinical isolates of *Candida* spp. Turk J Med Sci. 2020 Dec 17;50(8):2024-2031.*

*[2] Lim HJ, Shin JH, Kim MN, Yong D, Byun SA, Choi MJ, Lee SY, Won EJ, Kee SJ, Kim SH, Shin MG. Evaluation of Two Commercial Broth Microdilution Methods Using Different Interpretive Criteria for the Detection of Molecular Mechanisms of Acquired Azole and Echinocandin Resistance in Four Common *Candida* Species. Antimicrob Agents Chemother. 2020 Oct 20;64(11):e00740-20.*

[3] Ceballos-Garzon A, Holzapfel M, Welsch J, Mercer D. Identification and antifungal susceptibility patterns of reference yeast strains to novel and conventional agents: a comparative study using CLSI, EUCAST and Sensititre YeastOne methods. JAC Antimicrob Resist. 2025 Mar 19;7(2):dlaf040.

Line 59: The word "paediatric" should be corrected to "Pediatric"

The word has been corrected from "paediatric" to "pediatric" in the revised manuscript.

Line 104-110: description of the PUMCH hospital which is not the scope of the study and should be removed.

We appreciate the reviewer's suggestion. We agree that a lengthy description of PUMCH is beyond the scope of this study. Accordingly, we have shortened the section and retained only essential information to clarify the source of isolates.

Line 159: The word "denaturiona" should be corrected to "denaturation"

The word has been corrected from "denaturiona" to "denaturation" in the revised manuscript.

Line 185: The word "ematologic" should be replaced with "hematologic"

The word has been corrected from "ematologic" to "hematologic" in the revised manuscript.

Table 1 containing data for only 6 patients who have multiple isolates

Table 1 presents data only for the six patients who had multiple C. krusei isolates, to summarize cases with repeated isolates for clarity. We provide the complete isolates data in the supplementary materials (Supplement Table S1) .

Table 2 use the word "Antibiotic" should be replaced with "Antifungal"

The word has been corrected from "Antibiotic" to "Antifungal" in the revised manuscript.

Table 2: Remove the agreement/error data, which complicates interpretation without adding significant value.

While we understand that including agreement/error data may complicate interpretation, we believe that presenting the differences between CLSI and SYO MIC results is clinically valuable. Retaining this information helps clinicians better interpret antifungal susceptibility results and guide therapy, which we consider important for the manuscript.

Reviewer #2 (Comments for the Author):

COMMENTS TO AUTHORS

Lines 30, 90, 102: C. krusei must be written in italics.

We thank the reviewer for noticing this. All instances of C. krusei have been corrected to italics in the revised manuscript.

Line 43: The Fungal name is misspelled; it must be corrected

The fungal name has been corrected in the revised manuscript.

Lines 58, 61: We recommend that the authors write the whole name of the species at the beginning of the sentences.

We thank the reviewer for the suggestion. The full species name is already provided at the beginning of the paragraph; for readability and brevity, we have used the abbreviated form (C. krusei) in subsequent mentions.

Line 113, 119, 120, 123: The city name is missing from the Biomerieux reference.

The city name has been added to the Biomerieux references in the revised manuscript.

Line 133: Change C. parapsilosis to Candida parapsilosis and Candida krusei to C. krusei.

The species names have been corrected to Candida parapsilosis and C. krusei in the revised manuscript.

Line 154: If you developed the DNA extraction protocol, a detailed description of it should be provided. If it is based on a protocol by other authors, you should include the citation.

The genomic DNA was extracted using the Fungi Genomic DNA Extraction Kit (Solarbio Science & Technology, Beijing, China) according to the company's recommended protocols. We have revised the manuscript accordingly.

Line 166: The city name is missing from the ABI reference.

The city name has been added to the ABI references in the revised manuscript.

Line 185: "hematologic" is misspelled; it must be corrected.

The word has been corrected from "ematologic" to "hematologic" in the revised manuscript.

Line 187: We highly recommend that you erase "etc." and provide details of those 25 patients. Why did you decide to detail some of the clinical conditions and not all of them?

We provide the complete patient data in the supplementary materials (Supplement Table S1).

Line 241: Write "in blood samples, it WAS 0 %" instead of "in blood samples, it IS 0 %".

We have revised the sentence as suggested and changed "it is 0%" to "it was 0%".

Lines 319-321: We somehow disagree with this conclusion made by Ying Zhao and colleagues: "Bloodstream isolates are more likely to be subject to systemic antifungal therapy, particularly in hospitalized patients with invasive candidiasis, which may selectively FAVOR STRAINS THAT RETAIN SUSCEPTIBILITY". Isolates that have been exposed to antifungals and have survived in their presence tend to develop resistance, at least in *Candida albicans* isolates. Therefore, we would expect higher MIC values for bloodstream isolates rather than non-blood isolates. How can you explain this? Could you provide any reference to support your conclusion?

*We thank the reviewer for this important point. We agree that systemic antifungal exposure during candidemia can select for resistant strains in some settings; however, multiple lines of evidence indicate that non-blood (colonizing or device/sterile-site) isolates may also — and under some circumstances more often — demonstrate reduced susceptibility compared with contemporaneous bloodstream isolates. First, colonizing isolates that have been exposed to prolonged azole therapy frequently show higher rates of azole nonsusceptibility than initial blood isolates, as documented in a multicenter study of post-treatment colonizing *Candida* isolates [1]. Second, biofilm-associated and device-related isolates (which are more common among non-blood/sterile site specimens) display markedly increased antifungal tolerance compared with planktonic bloodstream isolates, a mechanism that can raise observed MICs and clinical resistance [2]. Third, surveillance and comparative studies have reported higher MICs or lower azole susceptibility among non-blood specimens for certain species (including reports from large surveillance programs), supporting that source-dependent differences exist and are clinically relevant [3-6].*

*Mechanistically, these observations can be explained by at least three, not-mutually-exclusive processes: (1) local/prior antifungal exposure at mucosal or device sites driving selection of resistant subpopulations before bloodstream invasion; (2) biofilm formation on catheters or tissue surfaces that increases phenotypic tolerance to antifungals; and (3) sampling/epidemiologic bias (different species distributions or patient populations at different sites). Reviews of *Candida* resistance and acquired mechanisms summarize how prior drug exposure and adaptive genomic changes produce stable resistance in clinical isolates [7-8].*

In view of these points, we believe the expectation that bloodstream isolates must always have higher MICs is not universally supported by the literature. Nevertheless, we acknowledge that both possibilities exist and that distinguishing true primary colonization from secondary seeding remains a limitation in single-center series. We have therefore revised the Discussion to (1) cite the above literature, (2) explain the plausible mechanisms for higher MICs in non-blood isolates (prior local exposure, biofilms, species mix), and (3) explicitly acknowledge the sampling-bias limitation and the need for prospective, paired (site + blood) studies to resolve this question definitively.

*[1] Jensen RH, Johansen HK, Søres LM, Lemming LE, Rosenvinge FS, Nielsen L, Olesen B, Kristensen L, Dzajic E, Astvad KM, Arendrup MC. Posttreatment Antifungal Resistance among Colonizing *Candida* Isolates in Candidemia Patients: Results from a Systematic Multicenter Study. *Antimicrob Agents Chemother.* 2015 Dec 28;60(3):1500-8. doi: 10.1128/AAC.01763-15.*

*[2] Silva S, Rodrigues CF, Araújo D, Rodrigues ME, Henriques M. *Candida* Species Biofilms' Antifungal Resistance. *J Fungi (Basel).* 2017 Feb 21;3(1):8.*

[3] Díaz-García J, Mesquida A, Gómez A, Machado M, Martín-Rabadán P, Alcalá L, Sánchez-Carrillo C, Reigadas E,

Vicente T, Muñoz P, Escribano P, Guinea J. Antifungal Susceptibility Testing Identifies the Abdominal Cavity as a Source of *Candida glabrata*-Resistant Isolates. *Antimicrob Agents Chemother*. 2021 Nov 17;65(12):e0124921.

[4] Pfaller MA, Diekema DJ, Gibbs DL, Newell VA, Nagy E, Dobiasova S, Rinaldi M, Barton R, Veselov A; Global Antifungal Surveillance Group. *Candida krusei*, a multidrug-resistant opportunistic fungal pathogen: geographic and temporal trends from the ARTEMIS DISK Antifungal Surveillance Program, 2001 to 2005. *J Clin Microbiol*. 2008 Feb;46(2):515-21.

[5] Badiie P, Choopanizadeh M, Moghadam AG, Nasab AH, Jafarian H, Shamsizadeh A, Soltani J. Antifungal susceptibility patterns of colonized *Candida* species isolates from immunocompromised pediatric patients in five university hospitals. *Iran J Microbiol*. 2017 Dec;9(6):363-371.

[6] Singh R, Kaur M, Chakrabarti A, Shankarnarayan SA, Rudramurthy SM. Biofilm formation by *Candida auris* isolated from colonising sites and candidemia cases. *Mycoses*. 2019 Aug;62(8):706-709.

[7] Bhattacharya S, Sae-Tia S, Fries BC. Candidiasis and Mechanisms of Antifungal Resistance. *Antibiotics (Basel)*. 2020 Jun 9;9(6):312.

[8] Morais Vasconcelos Oliveira J, Conceição Oliver J, Latércia Tranches Dias A, Barbosa Padovan AC, Siqueira Caixeta E, Caixeta Franco Ariosa M. Detection of ERG11 Overexpression in *Candida albicans* isolates from environmental sources and clinical isolates treated with inhibitory and subinhibitory concentrations of fluconazole. *Mycoses*. 2021 Feb;64(2):220-227.

Line 328: *Candida* must be written in italics.

It has been corrected to italics in the revised manuscript.

Line 382: spp. must not be written in italics.

It has been corrected in the revised manuscript.

Lines 408-410: It should be very interesting to know this clinical data; this would make it possible to determine whether the resistance to some antifungal compounds is intrinsic or has been developed during antifungal exposure.

Thank you very much for this insightful comment. We fully agree that clinical data such as prior antifungal treatment, underlying diseases, and patient outcomes would provide valuable context for interpreting strain behavior. However, as this was a retrospective study based on the collection of isolates and available information, these clinical data were not systematically recorded and are therefore incomplete. We have clarified this limitation in the revised manuscript.

Line 419: "Susceptibility" is misspelled; it must be corrected.

We have corrected the word in the revised manuscript.

Line 452: The names of all microorganisms in references must be written in italics.

We have corrected all microorganisms' names in references in the revised manuscript.

Reviewer #1 (Comments for the Author):

The authors have adequately addressed all the points raised in the previous revision. A few additional minor points should be corrected as follows:

- Table 1: Define the abbreviation FUO.

We appreciate the reviewer's helpful comment. The abbreviation "FUO" (fever of unknown origin) has now been defined in Table 1 accordingly.

- Table 3: In the title, change "14 clusters found in microsatellite typing" to "Fourteen clusters found in microsatellite typing."

Thank you for your suggestion. We have revised the title of Table 3 to "Fourteen clusters found in microsatellite typing" as recommended.

Re: Spectrum02608-25R2 (Phenotypic and Genotypic Differences Between Bloodstream and Non-Blood Candida krusei Isolates: Implications for Invasiveness and Antifungal Susceptibility)

Dear Prof. Ying Zhao:

Your manuscript has been accepted, and I am forwarding it to the ASM production staff for publication. Your paper will first be checked to make sure all elements meet the technical requirements. ASM staff will contact you if anything needs to be revised before copyediting and production can begin. Otherwise, you will be notified when your proofs are ready to be viewed.

Sincerely,
Matthew Anderson
Editor
Microbiology Spectrum